# Finding of Novel Galactose Utilizing *Halomonas* sp. YK44 for Polyhydroxybutyrate (PHB) Production

**DOI:** 10.3390/polym14245407

**Published:** 2022-12-10

**Authors:** Hee Ju Jung, Su Hyun Kim, Do Hyun Cho, Byung Chan Kim, Shashi Kant Bhatia, Jongbok Lee, Jong-Min Jeon, Jeong-Jun Yoon, Yung-Hun Yang

**Affiliations:** 1Department of Biological Engineering, College of Engineering, Konkuk University, Seoul 05029, Republic of Korea; 2Institute for Ubiquitous Information Technology and Applications, Konkuk University, Seoul 05029, Republic of Korea; 3Department of Biological and Chemical Engineering, Hongik University, Sejong 30016, Republic of Korea; 4Green & Sustainable Materials R&D Department, Research Institute of Clean Manufacturing System, Korea Institute of Industrial Technology (KITECH), Cheonan 31056, Republic of Korea

**Keywords:** polyhydroxybutyrate 1, *Halomonas* sp. YK44 2, galactose 3

## Abstract

Polyhydroxybutyrate (PHB) is a biodegradable bioplastic with potential applications as an alternative to petroleum-based plastics. However, efficient PHB production remains difficult. The main cost of PHB production is attributed to carbon sources; hence, finding inexpensive sources is important. Galactose is a possible substrate for polyhydroxyalkanoate production as it is abundant in marine environments. Marine bacteria that produce PHB from galactose could be an effective resource that can be used for efficient PHB production. In this study, to identify a galactose utilizing PHB producer, we examined 16 *Halomonas* strains. We demonstrated that *Halomonas cerina* (*Halomonas* sp. YK44) has the highest growth and PHB production using a culture media containing 2% galactose, final 4% NaCl, and 0.1% yeast extract. These culture conditions yielded 8.98 g/L PHB (78.1% PHB content (*w*/*w*)). When galactose-containing red algae (*Eucheuma spinosum*) hydrolysates were used as a carbon source, 5.2 g/L PHB was produced with 1.425% galactose after treatment with activated carbon. Since high salt conditions can be used to avoid sterilization, we examined whether *Halomonas* sp. YK44 could produce PHB in non-sterilized conditions. Culture media in these conditions yielded 72.41% PHB content. Thus, *Halomonas* sp. YK44 is robust against contamination, allowing for long-term culture and economical PHB production.

## 1. Introduction

Plastic products are convenient in various fields such as food packaging, sterile medical uses, and construction. However, widespread plastic use leads to a huge amount of waste, which adversely affects the environment and humanity worldwide [1,2,3]. In particular, petroleum-based, non-degradable plastics cause environmental pollution, thus demonstrating the need for degradable plastic alternatives [4,5].

Polyhydroxyalkanoates (PHAs) are bioplastics that are potential alternatives to artificially synthesized plastics [6]. PHAs can be used for multiple applications such as biomedical devices, packaging, and coating materials. Poly(3-hydroxybutyrate) (PHB) is a PHA with properties similar to synthetic polymers [7]. PHB is produced by many microorganisms (*Halomonas* sp., *Ralstonia eutropha*, *Bacillus* sp., and *Pseudomonas* sp., etc.) [8,9], and several PHAs are produced by bacteria using commercial sugars [10,11]. PHB has a higher production cost than petrochemical plastics; hence, several researchers are studying low-cost PHA production by investigating readily available alternative and inexpensive carbon sources [12,13,14].

Marine biomasses such as large green algae, red algae, and brown macroalgae are promising because they present fast growth, large growth areas, and have low land, fertilizer, and water requirements [15]. In addition, these algae contain a large number of polysaccharides; hence, abundant monosaccharides can be derived from these species [16,17]. Among these, *Eucheuma spinosum*, red algae, grows in Southeast area and is used to produce agar and carrageenan [18]. Agar is a viscous complex polysaccharide constituting the cell wall of red algae and is a galactose polymer. Carrageenan is a linear polysaccharide composed of galactan with connected galactose residues [19]; 3,6-anhydrogalactose and sulfate ester groups are connected to the galactose unit, which cause structural deformation during acid treatment or heat treatment [20]. The ratio of 3,6-anhydrogalactose in *E. spinosum* is 56.2% galactose and 43.8% 3,6-anhydrogalactose [21,22,23,24].

The marine environment contains several strains that use galactose, suggesting that these could be used to generate galactose for PHB production [25,26,27,28]. *Halomonas* strains are present in salt-rich environments such as salt lakes, salt sand, salt soil, saline wells, solar salterns, saline wetlands, and marine environments [29,30]. *Halomonas* spp. can survive under high salinity [31,32], and their PHB production utilizing galactose has been organized into a table for comparison (Table 1) [25,26,27,28].

Many researchers examined PHB production using galactose in many marine strains (Table 1) [25,26,27,28], however, previous strains did not give significant titer for PHB production. As a result we tried to find out PHB producers with higher production capabilities from galactose and hypothesized that halotolerant bacteria could be applied to seaweed hydrolysates that have high salinity.

To realize this in this study, we identified *Halomonas* sp. YK44 as a potential PHB-producing species through utilizing galactose. Then, we examined the optimal culture media conditions for *Halomonas* sp. YK44 and tested its viability during PHB production. In addition, the *E.spinosum* hydrolysates were used to see if *Halomonas* could produce PHB using red algae that is abundant in marine environments [18].

## 2. Materials and Methods

### 2.1. Chemical Reagents

All chemicals used for cell culture were purchased from BD Difco Laboratories (Becton-Dickinson Franklin Lakes, NJ, USA). Reagents used for GC (Gas chromatography) and HPLC (High Performance Liquid Chromatography) analysis (chloroform, methanol, and other derivatization reagents) were purchased from Sigma-Aldrich (St. Louis, MO, USA). Red seaweed hydrolysate was prepared from *E. spinosum*, which was procured from the Korea Institute of Industrial Technology (Cheonan, KITECH, Korea). The lysate was prepared using dilute H_2_SO_4_ and an acid catalyst.

### 2.2. Sample Collection, Strain Isolation, and Phylogenetic Analysis

All sixteen strains were obtained from the National Marine Biodiversity Institute of Korea (Seochun, NMBIK, Korea). Each stock sample was suspended with distilled water and was spread on marine agar (MA; Difco Laboratories, Detroit, MI, USA) for isolation. The plates were incubated for two days at 30 °C. Colonies were isolated from each plate and cultured for one day in marine broth (MB; Difco Laboratories, Detroit, MI, USA). Stocks were prepared with 20% (*w*/*v*) glycerol and were stored at −81 °C until use.

The strain was identified at the species level using 16S rRNA. 16s rRNA was acquired from NMBIK, compared to those in the GenBank database of the NCBI using BLASTN tools, and used for making a phylogenetic tree utilizing MEGA 11 software, Molecular Evolutionary Genetics Analysis version 11 (Tamura, Stecher, and Kumar 2021).

### 2.3. Plate Assay for Strain Characterization

Plate assays were used to test the characteristics of the isolated strains [33]. An antibiotic susceptibility test was also performed via plate assay. The isolated strains were cultured for 2 days on MA with each antibiotic (100 µg/mL ampicillin, 25 µg/mL gentamicin, 50 µg/mL kanamycin, 100 µg/mL spectinomycin, and 25 µg/mL chloramphenicol). Colonies present after selection were considered resistant to the antibiotic.

### 2.4. Culture Conditions for PHA Synthesis

To identify which of the 16 *Halomonas* strains produces PHB, we cultivated 16 strains using MB containing 2% fructose. To optimize PHB production, *Halomonas* sp. YK44 was cultivated in the presence of various carbon and nitrogen sources. To determine which carbon source *Halomonas* uses, we added 2% glucose, xylose, galactose, glycerol, sucrose, fructose, or lactose to MB. Then, 0.1% yeast extract, NH_4_NO_3_, NH_4_CL, (NH_4_)HSO_4_, NH_4_NO_3_, and (NH_4_)_2_SO_4_ were used to determine the optimal nitrogen source. In addition, galactose (1–5% *w*/*v*) and yeast extract (0.1–1% *w*/*v*) were added to MB to optimize galactose and yeast extract concentrations. All experiments were performed in a shaking incubator at 30 °C (200 rpm) for 48 h. To determine optimal growth conditions, *Halomonas* sp. YK44 was cultured in MB with different concentrations of final NaCl (2–10% *w*/*v*) with 2% galactose and cultivated at 16 °C, 20 °C, 25 °C, 30 °C, 37 °C, and 42 °C. To confirm the optimal cultivation time for PHA production, *Halomonas* sp. YK44 was cultured in MB containing 2% galactose, final 4% NaCl, and 0.1% yeast extract at 25 °C for 96 h. The residual carbon source concentration was evaluated by HPLC using a PerkinElmer system equipped with a refractive index detector (Waltham, MA, USA) and an Aminex HPX-87H column (300 × 7.8 mm internal diameter)(Dublin, Ireland). The flow rate of the mobile phase containing 0.008 N sulfuric acid was constantly maintained at 0.6 mL/min. The oven temperature was set at 60 °C [34].

### 2.5. Analytical Methods

Using GC, PHB were quantified and characterized as previously described [34,35]. After culturing, the broth was centrifuged to collect cell pellets, which were subsequently washed twice with deionized water. After adjusting the amount of water to match the culture volume, 1 mL samples were collected in a glass vial for lyophilization. The dry weight of the lyophilized pellet was measured. Then, 1 mL chloroform and 1 mL 15% (*v*/*v*) H_2_SO_4_/85% methanol solution were added to the dried cell pellet. Methanolysis was conducted while heating for 2 h at 100 °C. After cooling to room temperature (25 °C), 1 mL of deionized water was added to the methyl ester solution, which was vortexed for approximately 5 s. The chloroform layer was carefully extracted and transferred to a microtube containing sodium sulfate anhydrous to remove any remaining water. Samples were filtered through a 0.22-μm Millex-GP syringe filter unit and 1 μL aliquots were injected into a gas chromatograph with split mode (1/10) (Young-lin 6500, Seoul, Korea), equipped with a fused silica capillary column (Agilent HP-FFAP, 30 m × 0.32 mm, i.d. 0.25 μm film) and a flame ionization detector (FID). The inlet temperature was 210 °C and helium carrier gas was supplied at 3 mL/min. The oven temperature was controlled following a gradient program of 0–5 min at 80 °C and 12–17 min at 220 °C. The FID temperature was maintained at 230 °C throughout the operation [34,36,37].

### 2.6. TEM Analysis

To conduct TEM analysis, 1 mL of cultured sample was collected, centrifuged, and mixed with Karnovsky’s fixative solution containing 2% glutaraldehyde. The sample was fixed with 1% osmium tetroxide in 0.05 M sodium cacodylate buffer. The cells were gradually dehydrated using an ascending ethanol gradient (50%, 70%, 95%, and 100% *v*/*v*). Propylene was used for the transition step. The sample was settled in different ratios of propylene oxide and Spurr’s resin (*v*/*v*) (1:1 and 1:2). After mixing with 100% Spurr’s resin, the sample solidified overnight at 70 °C in a dry oven. Then, the samples were cut with an ultramicrotome (LEICA, EM UC7, Wetzlar, Germany). The slices were placed on a grid for EF-TEM (Carl Zeiss, LIBRA 120, Oberkochen, Germany) with an accelerating voltage of 120 kV [34].

### 2.7. Polymer Extraction and Characterization

*Halomonas* sp. YK44 cells were cultivated in 40 mL MB medium with 2% galactose for 48 h at 25 °C. The cells were pelleted, washed twice with deionized water, and lyophilized. Approximately 20 mL chloroform was added to the lyophilized cells and PHB was extracted for 16 h at 60 °C. The chloroform layer with dissolved polymer was collected by centrifugation and filtered using Whatman No. 1 filter paper to remove cell debris. The chloroform was evaporated at room temperature, to make a PHB film. The film was analyzed by differential scanning calorimetry (DSC) and gel permeation chromatography (GPC) [35,38].

DSC was used to analyze the extracted PHB film. Analysis was conducted with a Discovery DSC instrument (TA Instrument, Delaware, USA) from −60 °C to 180 °C. The heating and cooling rate was 10 °C/min in an N_2_ atmosphere. PHB powder (Sigma-Aldrich) was used as a standard.

GPC analysis was performed using a GPC instrument (YL Chromass, Anyang, Korea) with a loop injector (Rheodyne 7725i), dual-head isocratic pump (YL9112), column oven (YL9131), column (Shodex, K-805, 8.0 I.D. × 300 mm), and RI detector (YL9170). For GPC analysis, 0.1 g PHB film was dissolved in chloroform and passed through a 0.2-μm syringe filter (Chromdisc, Hwaseong, Korea). Here, 60 μL of this solution (without air bubbles) was used as the injection volume. The flow rate of the chloroform mobile phase was 1 mL/min and was maintained at 35 °C. The molecular masses of 5000–2,000,000 Da were compared to polystyrene standards using YL-Clarity software (YL Chromass, Anyang, Korea).

### 2.8. PHB Production in E. spinosum Hydrolysate

*Halomonas* sp. YK44 was cultured with *E. spinosum* hydrolysate in MB. To determine the optimal culture growth conditions, *Halomonas* sp. YK44 was cultured in MB with different concentrations of *E. spinosum* hydrolysate (final galactose concentration: 0.475–1.9% *w*/*v*) and pH 6–10. DCW and PHB were measured after 2 days of cultivation at 30 °C while shaking at 200 rpm. The salinity of *E. spinosum* hydrolysate was analyzed using a digital salinity tester (ATAGO, Tokyo, Japan).

The *E. spinosum* hydrolysate was treated with 1 g/L activated carbon. Activated carbon treatment was performed by shaking for 5 min and centrifuging at 3700 rpm for 15 min. The treated hydrolysate was filtered using a 0.2-μm syringe filter (Chromdisc, Hwaseong, Korea).

### 2.9. PHB Production Reusing Non-Sterilized and Sterilized Medium for Comparison

With or without sterilization, *Halomonas* sp. YK44 was cultured in a high salinity MB medium with final 6% NaCl, 2% galactose and 0.1% yeast extract for 2 days at 25 °C and 200 rpm for comparing non-sterilized and sterilized media.

Culture medium reuse experiments were conducted by comparing culture media in sterilized medium with that in non-sterilized media. After centrifugation, the supernatant was collected and reused. We added 2% galactose and 0.1% yeast extract to the reused culture medium each cycle. It was done 10 cycles in total.

## 3. Results

### 3.1. Isolation and Characterization of Marine Strains for PHA Production

16 *Halomonas* strains were cultured in MB containing 2% fructose and analyzed for PHB production using GC. Among them, we found that *Halomonas* sp. YK44 produces the highest amount of PHB (11.5 g/L of DCW, 6.8 g/L *w*/*w* PHB; 58.8% PHB content) (Table 2, Figure 1A). Strain YK44 showed high 16s rRNA sequence similarity (97.93%) with *H. cerina* SP4 (red box) (Figure 1B) according to phylogenetic analysis. Given the high PHB peak, *Halomonas* sp. YK44 was selected for further study.

*Halomonas* sp. YK44 was assessed for hydrolase activity, antibiotic resistance, carbon and nitrogen source utility, growth temperature, and salt resistance. *Halomonas* sp. YK44 did not show any hydrolase activity. To evaluate antibiotic resistance, the isolated strain was cultured in MB containing each 100 µg/mL ampicillin, 25 µg/mL gentamicin, 50 µg/mL kanamycin, 100 µg/mL spectinomycin, and 25 µg/mL chloramphenicol. Strain YK44 showed the highest resistance on the ampicillin plate. The strain was also resistant to spectinomycin (Appendix A). When YK44 was tested for poly(3-hydroxybutyrate-co-3-hydroxyvalerate) (P(3HB-co-3HV) production from propionate, it produced only 0.43% 3HV.

To visualize PHA accumulation in *Halomonas* sp. YK44, the strain was cultured for 48 h at 25 °C in MB with 2% galactose, final 4% NaCl, and 0.1% yeast extract. Then, TEM analysis was conducted. TEM showed PHB granule accumulation in the cells after 48 h cultivation (Figure 1C). The PHB granules completely filled the cytoplasm showing *Halomonas* sp. YK44 produces PHB.

### 3.2. Evaluating Carbon and Nitrogen Sources for PHB Production

Among the tested carbon sources (2% glucose, xylose, fructose, galactose, sucrose, glycerol), *Halomonas* sp. YK44 showed the best growth with sucrose but showed higher PHB production in the presence of galactose (Figure 2A; Table 1). To optimize the galactose concentration, we compared PHB production with different galactose concentrations (1–5% *w*/*v*). Growth and PHB production were the highest at 2% galactose; hence, this percentage was used for further analysis (Figure 2B).

Multiple nitrogen sources, such as 0.1% yeast extract, NH_4_NO_3_, NH_4_CL, (NH_4_)HSO_4_, NH_4_NO_3_, and (NH_4_)_2_SO_4_, were added to MB to determine the optimal nitrogen source. *Halomonas* sp. YK44 showed the most growth and PHB production with 0.1% yeast extract, followed by (NH_4_)_2_SO_4_ (Figure 2C). To optimize the concentration of yeast extract, we compared PHB production in MB with different yeast extract concentrations (0.1–1% *w*/*v*). However, additional yeast extract did not change PHB production. Since PHB production was similar in MB containing 0.1% and 1% yeast extract, we decided to use 0.1% yeast extract to facilitate economical PHB production (Figure 2D).

### 3.3. Examination of Culture Temperature, Time, and NaCl Concentration

To determine the optimum culture conditions, we first examined different culture temperatures. When cultured at 16, 20, 25, 30, 37, and 42 °C, YK44 showed the best growth and PHB production at 25 °C (Figure 3A). Other studies mainly cultured *Halomonas* strains at 30 °C and 37 °C [39,40,41]. YK44 can grow and produce PHB at 25 °C similar to room temperature.

Since *Halomonas* sp. YK44 is halotolerant and can grow under high NaCl conditions, we examined how various salt concentrations affected PHB production (Figure 3B). Growth and PHB production were highest under final 4% NaCl, though there was no difference of more than 2 g/L from 2% to final 8% NaCl.

To determine the optimal *Halomonas* sp. YK44 culture time, growth in the optimized medium was monitored every 24 h for 4 days. The highest growth and PHB production were observed after 48 h (Figure 3C), with PHB production reaching 8.98 g/L PHB (75.11% PHB content *w*/*w*) and 11.6 g/L DCW. Culturing for more than 48 h did not increase PHB production, likely because most of the sugar was consumed. Finally, 20 g/L of galactose was completely consumed after 72 h.

### 3.4. Chemical Properties and Molecular Weight of PHB from Halomonas sp. YK44

PHB film was extracted from 40 mL of optimized cultivation media after 48 h cultivation at optimal culture conditions (25 °C in MB containing 2% galactose, final 4% NaCl, and 0.1% yeast extract) (Appendix A). The melting and crystallization temperatures of the extracted PHB film were T_m_ = 170.57 °C and T_c_ = 126.60 °C, respectively. These values were similar to those of authentic PHB film: T_m_ = 176.55 °C and T_c_ = 126.29 °C (Appendix A). Further, the T_m_ and T_c_ values were similar to those of authentic films and PHB films derived from other PHB producer strains [42,43,44].

To characterize the number average molecular weight (M_w_), the weight average molecular weight (M_n_), and the polydispersity index (PDI = M_w_/M_n_) of PHB extracted from *Halomonas* sp. YK44, we performed a GPC analysis. The M_w_, M_n_, and PDI of extracted PHB were 7.92 × 10^5^, 6.90 × 10^5^, and 1.48, respectively. GPC analysis revealed that strain YK44 produced higher molecular weight polymers than authentic PHB films (M_w_ = 4.29 × 10^5^, M_n_ = 3.19 × 10^5^, PDI = 1.34) (Appendix A). A high molecular weight polymer indicates high PHB quality, which is advantageous for industrial applications. Reduced molar masses during extraction are likely caused by damage to the granules. Compared with authentic PHB, intracellular PHB has low polydispersity; PDI is narrower than authentic PHB film. The narrow dispersity means homogeneous polymers [45,46].

### 3.5. PHB Production using E. spinosum Hydrolysates

To determine possible applications of *Halomonas* sp. YK44, cells were cultured in MB containing *E. spinosum* hydrolysate. The salinity of *E. spinosum* hydrolysates was approximately 2.6% and contained 3.8% galactose. To optimize the pH of *E. spinosum* hydrolysate, we compared PHB production capacity in the MB medium at different pH values. Growth and PHB production were the highest at pH 10 (Figure 4A). Further, growth was suppressed at galactose concentrations higher than 0.95%, possibly due to the influence of inhibitors. The highest cell growth and PHB production were obtained with 0.95% galactose concentration from *E. spinosum* hydrolysate in MB. Culturing in these conditions returned 4.9 g/L DCW and 27.2% PHB content (Figure 4B).

Activated carbon was used to remove impurities from the hydrolysate as activated carbon treatment removes toxic substances that may be released during hydrolysis [47,48]. *Halomonas* sp. YK44 showed optimal growth with 0.95% galactose but activated carbon treatment led to growth in media containing 1.9% galactose. Moreover, 5.2 g/L PHB was produced under 1.425% galactose culture condition (Figure 4C), which was 3.92-fold higher than the culture without activated carbon. These results suggest that activated carbon treatment increases hydrolysate utilization.

### 3.6. PHB Production in Reusing High Saline Non-Sterilized Medium and Sterilized Medium

To examine possible applications of *Halomonas* sp. YK44, we investigated whether this strain could be cultured in non-sterilized high-salt media without contamination and whether these culture conditions would change PHB production [49,50]. In sterilized or non-sterilized MB with final 6% NaCl and 2% galactose, *Halomonas* sp. YK44 produced PHB with or without sterilization. In non-sterilized samples, the highest amount of PHB was 7.24 g/L from 10.00 g/L DCW (72.41% PHB content *w*/*w*) in non-sterilized samples. This was comparable with results for sterilized media, which yielded 7.13 g/L PHB from 9.50 g/L DCW (75.02% PHB content *w*/*w*) (Figure 5). Overall, these results indicate that *Halomonas* sp. YK44 produces PHB under non-sterilized conditions like sterilized conditions.

When comparing sterilized and non-sterile media for reperirive use of media, there is no significant difference in growth or PHB production, indicating that *Halomonas* sp. YK44 could be cultured in non-sterilized medium without contamination. Accumulated data over 10 cycles indicated that PHB production showed a difference of approximately 1.8 g/L, but this difference was not significant considering that it is an accumulation of PHB production (Figure 5). Thus, *Halomonas* sp. YK44 shows high robustness to contamination, which will facilitate PHB production for industrial applications.

## 4. Discussion

As *Halomonas* spp. can usually survive and produce PHB at high salinity, it is possible to utilize open non-sterile fermentation in a high salinity medium without antibiotics treatment or sterilization [31,32]. In addition, as this *Halomonas* strain is a benign native PHB producer, it is free of any environmental issues such as the law of genetically modified objects or pathogen issues. Since *Halomonas* sp. YK44 is salt-tolerant and uses galactose, this strain can be used to produce PHB by utilizing seaweed hydrolysate like *Euchenoma spinosum* hydrolysate resulting in reduced PHB production cost. Our results suggest that not only can red algae be used to produce PHB, but also that it allows microorganisms to grow better when hydrolysates are treated with activated carbon, as shown in other papers [51,52].

In conclusion, we determined that *Halomonas* sp. YK44 strain produces more PHB using galactose as a carbon source among 16 *Halomonas* strains. According to Figure 3B, as *Halomonas* sp. YK44 is salt-tolerant and uses galactose, it can be used to produce PHB utilizing seaweed hydrolysate. PHB was produced using *E. spinosum* hydrolysates that have abundant galactose with a salinity of 2.6% and production increased after activated carbon treatment. Furthermore, we reused the media to compare PHB productivity with sterilized and non-sterilized high-salt media to determine that *Halomonas* is resistant to contamination in high salinity conditions. Total PHB production showed no significant difference between cultures in sterilized or non-sterilized high-salinity culture media. Thus, cost-effective application will be possible even if PHB is produced using by-products without sterilization.

In *Halomonas* sp. YK44, PHB was effectively produced using various sugar components, especially galactose. We optimized culture media conditions to produce PHB in MB media with 2% galactose, 0.1% yeast extract, and final 4% NaCl concentration under 25 °C for 48 h. Under optimized PHB production conditions, the strain produced 8.98 g/L PHB from 11.6 g/L DCW (78.11% PHB content). *Halomonas* sp. YK44 showed about 50% more production than the maximum amount of the previously reported paper [25,26,27,28]. Similar PHB content was produced in MB medium containing final 2–8% NaCl. This suggests that *Halomonas* sp. YK44 is salt-tolerant similar to other *Halomonas* spp. [53,54]. Since PHB was produced at a low temperature (25 °C), it may be possible to produce only by shaking at room temperature. For PHB production and cultivation using *E. spinosum* hydrolysate, 1.425 g/L galactose, and activated carbon, the yield was 5.2 g/L PHB, which was the largest amount we produced using *E. spinosum* hydrolysates. In MB media with final 6% NaCl and 2% galactose without antibiotics and sterilization, PHB production was 4.2 g/L PHB from 7.3 g/L DCW.

## Figures and Tables

**Figure 1 polymers-14-05407-f001:**
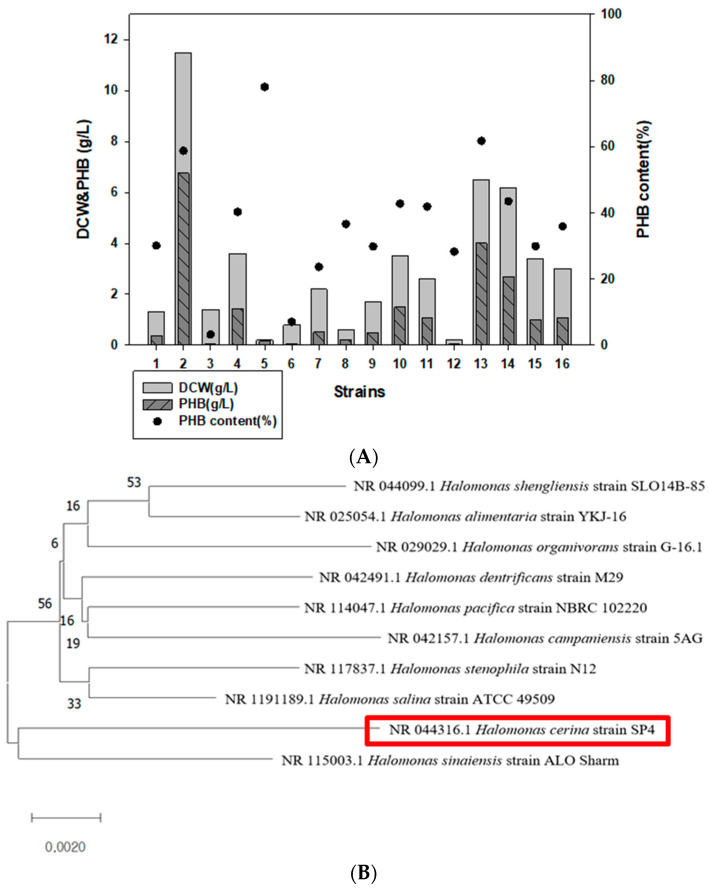
Identification of PHB-producing strains that produce PHB. (**A**) GC analysis comparison of DCW and PHB production in 16 *Halomonas* strains. (**B**) Phylogenetic analysis using 16S rRNA sequencing of *Halomonas* sp. YK44. (**C**) TEM image of *Halomonas* sp. YK44 after being cultured for 48 h at 25 °C in MB medium with 2% galactose, final 4% NaCl, and 0.1% yeast extract (Magnification: Right—16,300×, Left—25,000×).

**Figure 2 polymers-14-05407-f002:**
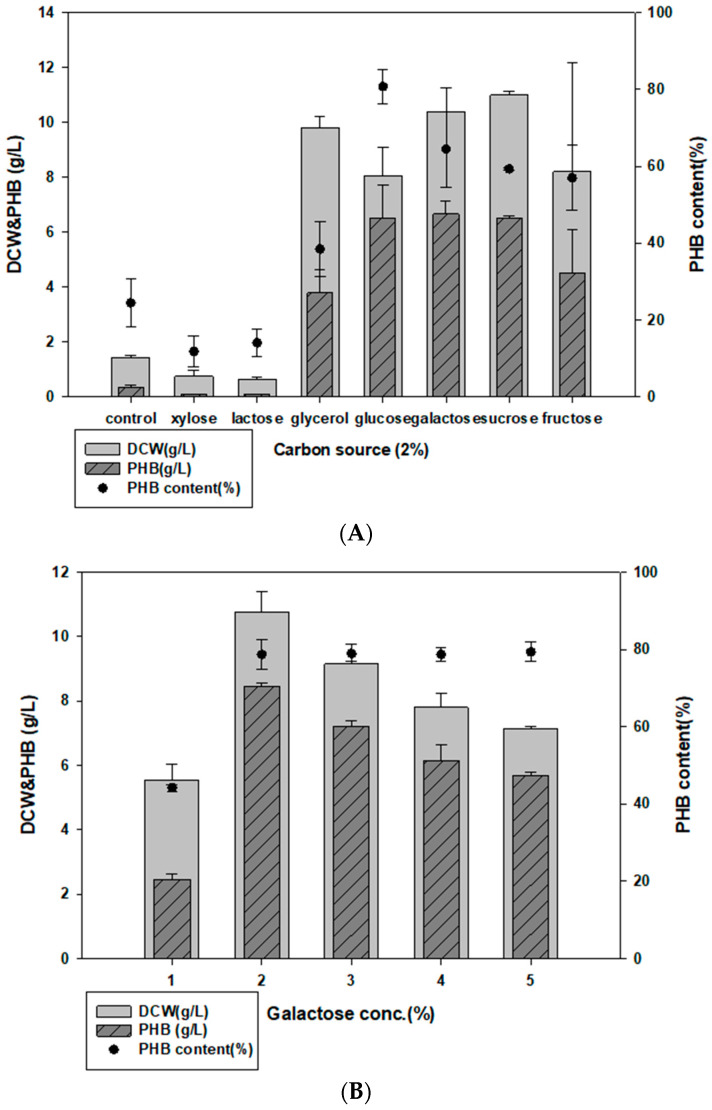
Optimized carbon and nitrogen sources for PHB production. (**A**) Comparison of PHB production in MB medium with 2% (*w*/*v*) additional carbon sources. (**B**) Effect of galactose concentration on PHB production. (**C**) Comparison of PHB production in MB medium with 0.1% (*w*/*v*) additional nitrogen sources. (**D**) Effect of yeast extract concentration on PHB production in MB.

**Figure 3 polymers-14-05407-f003:**
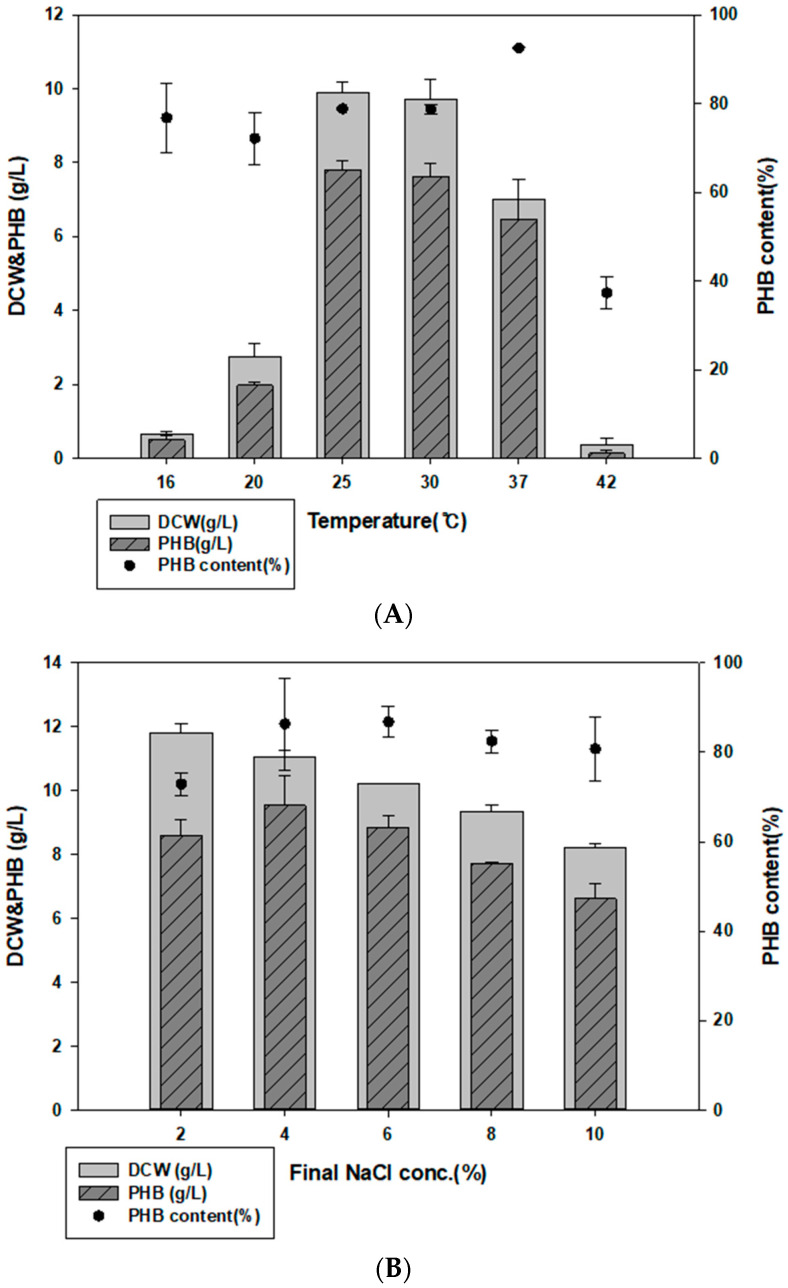
Optimized culture conditions for PHB production. (**A**) Effect of NaCl on PHB production in MB containing 2% (*w*/*v*) galactose. (**B**) Effect of temperature on PHB production in MB with 2% (*w*/*v*) galactose. (**C**) Optimized cultivation time for PHB production at 25 °C in MB with 2% galactose, final 4% NaCl, and 0.1% yeast extract.

**Figure 4 polymers-14-05407-f004:**
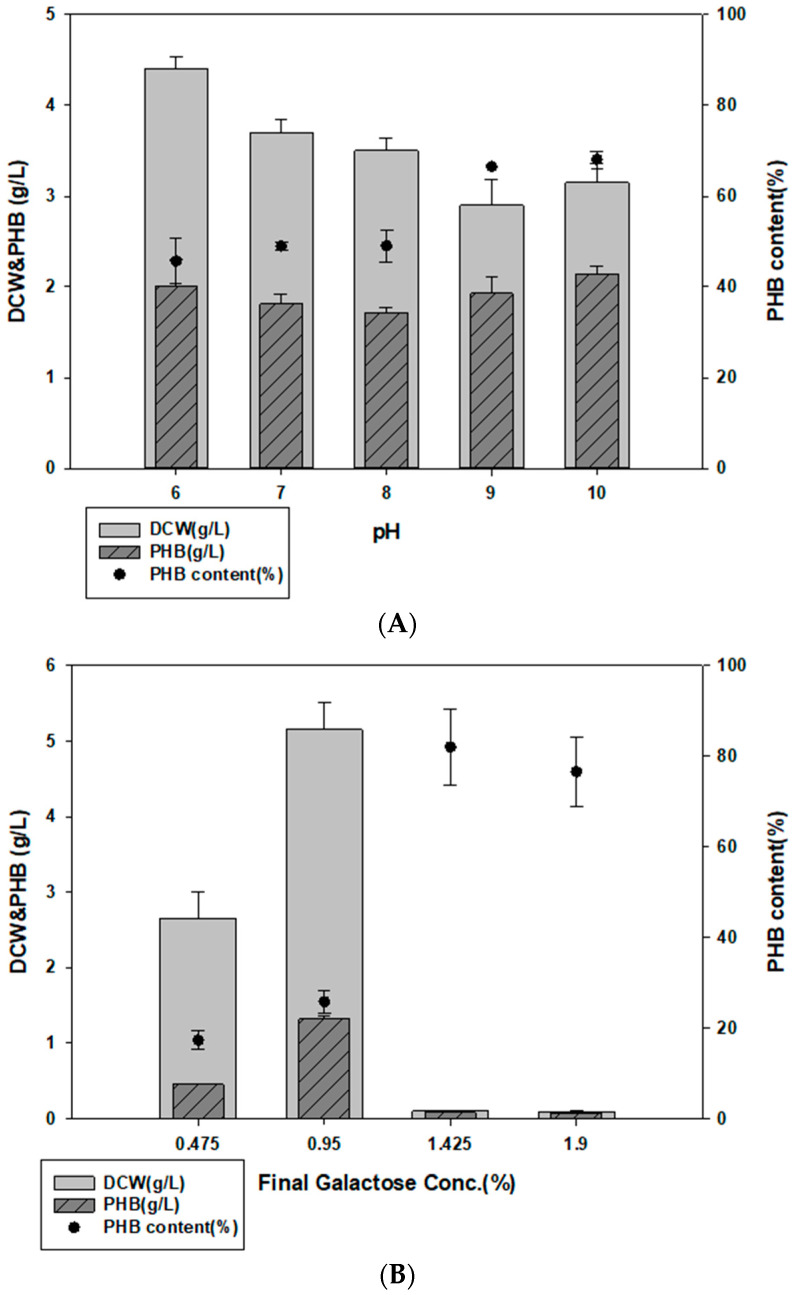
PHB production with *E. spinosum* hydrolysate and optimal pH. (**A**) Effect of pH on PHB production in MB with 1% galactose in *E. spinosum* hydrolysate. (**B**) DCW and PHB production after culture with *E. spinosum* hydrolysate (**C**) DCW and PHB production after culture with *E. spinosum* hydrolysate and activated carbon.

**Figure 5 polymers-14-05407-f005:**
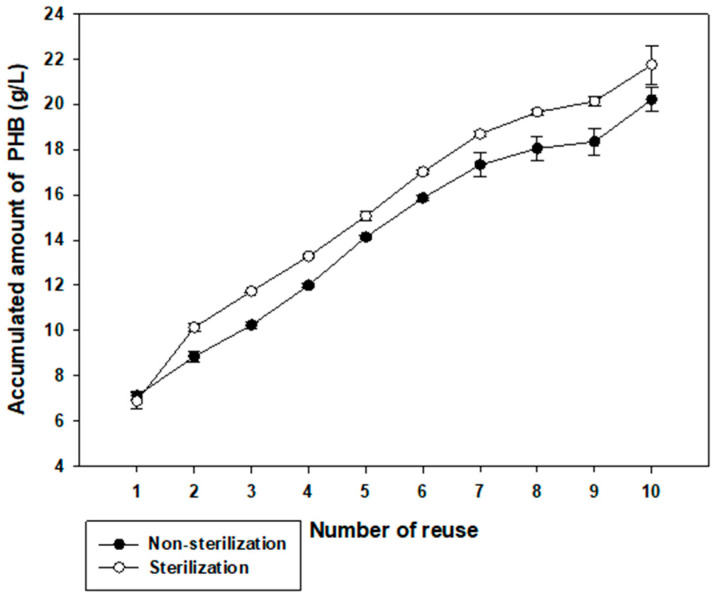
Effect of reused sterilized and non-sterilized high salinity medium on total PHB production. Total PHB production in non-sterilized high salinity medium with 2% galactose was compared to production in sterilized high salinity medium with 2% galactose.

**Table 1 polymers-14-05407-t001:** PHA synthesis using galactose.

Microorganism	Carbon Source	DCW (g/L)	PHB Content (wt%)	PHB (g/L)	Culture	Reference
*Halomonas halophila*	Galactose	4.22 ± 0.10	80.7 ± 2.0	3.41 ± 0.12	Batch	[27]
*Halomonas salina*	Galactose	0.97 ± 0.03	12.3 ± 0.48	0.12 ± 0.01	Batch	[25]
*Halomonas organivorans*	Galactose	5.80 ± 0.22	90.55 ± 4.08	5.61 ± 0.01	Batch	[25]
*Bacillus* sp. 112A	Galactose	1.02	35.50	0.879	Batch	[26]
*Halomonas* sp. SF2003	Galactose	3.16	39	1.23	Batch	[28]

**Table 2 polymers-14-05407-t002:** *Halomonas* strains used in this study.

	Similar Species	Strain	Isolated Temperature	Resource Number
1	*Halomonas alkalicola*	DH-10	25 °C	MABIK MI00000003
2	*Halomonas cerina*	YK44	20 °C	MABIK MI00000284
3	*Halomonas sulfidaeris*	J05-14M-11R	20 °C	MABIK MI00000306
4	*Halomonas fontilapidosi*	O12	25 °C	MABIK MI00000338
5	*Halomonas gomseomensis*	CJCa107	25 °C	MABIK MI00000370
6	*Halomonas arcis*	CJCbj078	25 °C	MABIK MI00000396
7	*Halomonas janggokensis*	CJCbj082	25 °C	MABIK MI00000412
8	*Halomonas salicampi*	CJCbj041	25 °C	MABIK MI00000417
9	*Halomonas lutescens*	CJCbj058	25 °C	MABIK MI00000427
10	*Halomonas fontilapidosi*	MEBiC12169	25 °C	MABIK MI00005505
11	*Halomonas campaniensis*	S510	25 °C	MABIK MI00005561
12	*Halomonas saccharevitans*	MJ005	25 °C	MABIK MI00005664
13	*Halomonas shengliensis*	MEBiC12098	25 °C	MABIK MI00005446
14	*Halomonas denitrificans*	MEBiC13328	25 °C	MABIK MI00005839
15	*Halomonas aestuarii*	MEBiC13369	25 °C	MABIK MI00005871
16	*Halomonas lutea*	15A021	27 °C	MABIK MI00005926

## Data Availability

All data related to the study are provided with the manuscript and its associated files.

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
