# Peer review of "Finding of Novel Galactose Utilizing *Halomonas* sp. YK44 for Polyhydroxybutyrate (PHB) Production"

_polymers, 2022, doi:10.3390/polym14245407_

Round 1
Reviewer 1 Report
we found some statements in the results that should be put into the methods.

Author Response
Response to Reviewer 1 Comments
R1Q1: Page 3, line91-92; “The strain was identified at the species level using 16S rRNA. 16s rRNA was acquired from NMBIK” Please explain the detail of methods?
R1A1: As NMBIK stood for National Marine Biodiversity Institute of Korea, we explained it in the manuscript. We got the strains and 16S rRNA information form marine biobank(https://www.mbris.kr/biobank/main/index.do) and confirmed it by 16S rRNA sequencing.
R1Q2: Page 3, line100; “Culture conditions for PHA synthesis” source of reference?
R1A2: As reviewer mentioned, we added references.
R1Q3: Page 5, line201-202; “To identify PHB producing strains, 16 Halomonas strains were examined. Each strain were cultivated in MB medium containing 2% fructose.” put this statement in Methods.
R1A3: As reviewer suggested, we moved this statement into Methods.
R1Q4: Page 8, line237-239; “To optimize PHB production, Halomonas sp. YK44 was cultivated in the presence of various carbon and nitrogen sources. First, 2% glucose, xylose, fructose, galactose, su- crose, glycerol, or lactose was added to MB.” put this statement in Methods.
R1A4: As reviewer suggested, we moved this statement into Methods.
Reviewer 2 Report
“Finding of novel galactose utilizing Halomonas sp. YK44 for Polyhydroxybutyr- 2 ate (PHB) production” is quite novel and interesting study. I appreciate the hard work by the researchers for the study, however there are some suggestions below for acceptance.
As a general review, I would also like to mention that, although the language quality is sufficient, it is hard to follow the paper as a whole so I recommend a check from a native speaker.
Page 2, Lines 72-73: “Considering these strains are robust to contamination and salt conditions, Halomonas sp. YK44 is promising for developing an open non-sterile fermentation process to reduce the cost of PHB production.” Please reformulate this sentence. It is not clear, it must be revised. So is the decision for developing open non-sterile fermentation process only related with the robustness?
Page 2, Lines 66-74: The paragraph starting with “Although we previously examined galactose.. “ should be rewritten. Please make it in passive voice and also reformulate to highlight the novelty of this work.
Page 5, Lines 184-197: I think this section is left here by mistake. Please remove, it is part of the submission guidelines possibly.
“The Materials and Methods should be described with sufficient details to allow others to replicate and build on the published results. Please note that the publication of your manuscript implicates that you must make all materials, data, computer code, and protocols associated with the publication available to readers. Please disclose at the submission stage any restrictions on the availability of materials or information. New methods and protocols should be described in detail while well-established methods can be briefly described and appropriately cited. ….”
Page 13, Lines 349-352: Same comment.
“Authors should discuss the results and how they can be interpreted from the per-
spective of previous studies and of the working hypotheses. The findings and their implications should be discussed in the broadest context possible. Future research directions may also be highlighted”
Figures
The figure resolutions are very low. Please redraw the figures. Please also make the axis data visible and increase the overall visual high quality. (Particularly for Figure 2a and 3c.) Please make the figure punto and sizes consistent.
It would be nice if you can add the figures for the PHB films produced.
Page 11, Lines 196-207: “The Mw, Mn, and PDI of extracted PHB were 7.92 × 105, 6.90 × 105, and 1.15, respectively.” Is it 1.15?
Discussion is so limited. It must be added.
Conclusion?
Author Response
Response to Reviewer 2 Comments
R2Q1: As a general review, I would also like to mention that, although the language quality is sufficient, it is hard to follow the paper as a whole so I recommend a check from a native speaker.
R2A1: As reviewer suggested, our manuscript was checked by native speaker again.
R2Q2: Page 2, Lines 72-73: “Considering these strains are robust to contamination and salt conditions, Halomonas sp. YK44 is promising for developing an open non-sterile fermentation process to reduce the cost of PHB production.” Please reformulate this sentence. It is not clear; it must be revised. So is the decision for developing open non-sterile fermentation process only related with the robustness?
R2A2: As reviewer suggested, we reformulated this sentence. As Halomonas usually could survie and produce PHB at high salinity, it is possible to open non-sterile fermentation in a high salinity medium without antibiotics treatment or sterilization. As this Halomonas is benign native PHB producer, it is free of any environmental issue like law of genetically modified object or pathogen issues. We tried to add it in the manuscript with references.
R2Q3: Page 2, Lines 66-74: The paragraph starting with “Although we previously examined galactose. “should be rewritten. Please make it in passive voice and also reformulate to highlight the novelty of this work.
R2A3: As reviewer suggested, we wrote sentence again and tried to emphasize the novelty of this work.
R2Q4: Page 5, Lines 184-197: I think this section is left here by mistake. Please remove, it is part of the submission guidelines possibly. “The Materials and Methods should be described with sufficient details to allow others to replicate and build on the published results. Please note that the publication of your manuscript implicates that you must make all materials, data, computer code, and protocols associated with the publication available to readers. Please disclose at the submission stage any restrictions on the availability of materials or information. New methods and protocols should be described in detail while well-established methods can be briefly described and appropriately cited. ….”
R2A4: We thank reviewer for pointing out of our mistake. We deleted this part.
R2Q5: Page 13, Lines 349-352: Same comment.
“Authors should discuss the results and how they can be interpreted from the per-
spective of previous studies and of the working hypotheses. The findings and their implications should be discussed in the broadest context possible. Future research directions may also be highlighted”
R2A5: We thank reviewer for pointing out of our mistake. We deleted this part.
R2Q6: The figure resolutions are very low. Please redraw the figures. Please also make the axis data visible and increase the overall visual high quality. (Particularly for Figure 2a and 3c.) Please make the figure punto and sizes consistent.
R2A6: As reviewer suggested, we drew the figures again and used bold font for making the axis data visible. We also made the figure size consistent.
R2Q7: It would be nice if you can add the figures for the PHB films produced.
R2A7: As reviewer suggested, we added PHB film produced at Supplemenrary figure. (Supplementary figure 2)
R2Q8: Page 11, Lines 196-207: “The Mw, Mn, and PDI of extracted PHB were 7.92 × 105, 6.90 × 105, and 1.15, respectively.” Is it 1.15?
R2A8: We recalculated PDI by dividing (7.92 × 105) with (6.90 × 105) resulting in 1.48. The incorrectly calculated answer was erased and replaced with the correct answer. We appreciate reviewer’s point.
R2Q9: Discussion is so limited. It must be added.
R2A9: As reviewer suggested, we added some contents about discussion.
R2Q10: Conclusion?
R2A10: As reviewer suggested, we wrote the conclusion in the Discussion part.
Round 2
Reviewer 2 Report
Thank you for the revisions. They look sufficient for acceptance.